# Genetic Polymorphisms in the 3′-Untranslated Regions of *SMAD5, FN3KRP*, and *RUNX-1* Are Associated with Recurrent Pregnancy Loss

**DOI:** 10.3390/biomedicines10071481

**Published:** 2022-06-22

**Authors:** Min-Jung Kwon, Ji-Hyang Kim, Jeong-Yong Lee, Eun-Ju Ko, Hyeon-Woo Park, Ji-Eun Shin, Eun-Hee Ahn, Nam-Keun Kim

**Affiliations:** 1Department of Biomedical Science, College of Life Science, CHA University, Seongnam 13488, Korea; 0906sally@naver.com (M.-J.K.); smilee3625@naver.com (J.-Y.L.); ejko05@naver.com (E.-J.K.); aabb1114@naver.com (H.-W.P.); 2Department of Obstetrics and Gynecology, CHA Bundang Medical Center, School of Medicine, CHA University, Seongnam 13496, Korea; bin0902@chamc.co.kr (J.-H.K.); 1219annie@cha.ac.kr (J.-E.S.)

**Keywords:** *SMAD5*, FN3KRP, RUNX-1, polymorphism, recurrent pregnancy loss

## Abstract

Recurrent pregnancy loss (RPL) is typically defined as two or more consecutive pregnancy losses prior to 20 weeks of gestation. Although the causes of idiopathic RPL are not completely understood, vascular development and glucose concentration were reported to correlate with the pregnancy loss. The TGF-β signaling pathway which plays a significant role in pregnancy is activated by the interaction between high glucose and SMAD signaling and affects the vascular cells. *SMAD5* and *RUNX-1* are involved in the TGF-β signaling pathway and contribute to advanced glycation end products (AGEs) production and vascular development. *FN3KRP*, a newly described gene, is also associated with vascular diseases and suggested to relate to AGEs. Therefore, in the present study, we investigated associations between RPL risk and genetic polymorphisms of *SMAD5, FN3KRP*, and *RUNX-1* in 388 women with RPL and 280 healthy control women of Korean ethnicity. Participants were genotyped using real-time polymerase chain reaction and restriction fragment length polymorphism assay to determine the frequency of *SMAD5* rs10515478 C>G, *FN3KRP* rs1046875 G>A, and *RUNX-1* rs15285 G>A polymorphisms. We found that women with RPL had lower likelihoods of the *FN3KRP* rs1046875 AA genotype (adjusted odds ratio (AOR), 0.553; *p* = 0.010) and recessive model (AOR, 0.631; *p* = 0.017). Furthermore, combination analysis showed that *SMAD5* rs10515478 C>G and *FN3KRP* rs1046875 G>A mutant alleles were together associated with reduced RPL risk. These findings suggest that the *FN3KRP* rs1046875 G>A polymorphism has a significant role on the prevalence of RPL in Korean women. Considering that it is the first study indicating a significant association between *FN3KRP* and pregnancy disease, RPL, our results suggest the need for further investigation of the role of *FN3KRP* in pregnancy loss.

## 1. Introduction

Recurrent pregnancy loss (RPL) is a relatively common condition defined as more than two clinical pregnancy losses prior to 20 gestational weeks [1]. RPL occurs in 1–5% of reproductive-aged women worldwide, but its exact cause is unknown in approximately half of cases [2,3]. While possible causes are suggested, RPL has been understudied, with multiple factors potentially involved, such as genetic factors, environmental factors and certain organic diseases [4]. Gestational diabetes, which is a common complication of pregnancy, is one factor that increases the risk of spontaneous abortion. Its main phenotype, high blood sugar, can induce widespread damage to the body including blood vessel dysfunction, which can contribute to infertility [5]. Adequate vascular function for carrying nutrients is essential for uterine and embryo development and successful birth [6,7]; when the vascular complications of diabetes occur, several problems such as preeclampsia, early birth and miscarriage can result [8]. In addition, high blood sugar contributes to the formation of advanced glycation end-products (AGEs), which are toxic metabolic products of lipids, nucleic acids, and proteins formed by non-enzymatic reactions with sugars [9]. Elevated levels of AGEs are associated with severe health issues such as cancer and vascular diseases and may also play a role in infertility [4].

The transforming growth factor-beta (TGF-β) signaling pathway, which is important for embryo development, is initiated by high glucose interacting with SMAD signaling [10] and stimulates AGE activation by interconnection of *SMAD5* and TGF-β1 [11]. Runt-related transcription factors (RUNXs) are also involved in the TGF-β signaling pathway, working synergistically with SMADs [12,13]. In particular, *RUNX-1* plays a similar role as *SMAD5* in mouse embryo, contributing to vascular system development [12,14,15,16], and was found to promote the expression of endothelial-specific molecules for vascular formation in the mouse yolk sac [17]. Moreover, *RUNX-1* is known as a glucose-related gene and suggested to mediate glucose-related disease [18,19,20]. Additionally, *SMAD5* and *RUNX-1* are target genes of miR-27a, a promoter of angiogenesis and glucose metabolism [21,22,23]. However, genetic variants of *SMAD5* and *RUNX-1* were not yet reported in pregnancy loss while there are reports about the significant polymorphisms in vascular disease and platelet dysfunction [24,25].

Fructosamine-3 kinase-related protein (*FN3KRP*) is a newly described gene that shares 65% similarity in structure with fructosamine-3-kinase (*FN3K*), which contributes to glucose metabolism by preventing the formation of AGEs [26,27,28,29]. Although it is not clear whether *FN3KRP* functions similarly to *FN3K*, a recent study indicates that a genetic polymorphism of *FN3KRP* is associated with glucose products [30]. In particular, *FN3KRP* rs1046875 G>A protects against cardiovascular diseases through HbA1c, glycated hemoglobin, and affects the binding affinity of miR-34a [31] which acts as a suppressor of angiogenesis and glucose metabolism [32,33]. Although the mechanism of HbA1c has to be elucidated, its increased level has been reported in patients with cardiometabolic diseases and it has positive correlation with the platelet dysfunction in diabetic mellitus [34,35,36]. As a precursor of hemoglobin-AGE, higher HbA1c concentration is also associated with various diseases such as vascular diseases, diabetes and pregnancy disorders [34,35,36,37]. Thus, *FN3KRP* rs1046875 G>A has a potential probability to play a significant role in pregnancy in relation to HbA1c.

Recent studies showed that either vascular function or glucose-related genes influence RPL risk [38,39,40]. In addition, we previously found that pregnancy disorders are associated with genetic polymorphisms in 3′-untranslated regions (3′-UTRs) [41,42]. 3′-UTR regulates gene expression by binding with microRNA (miRNA), thereby modifying the stability and function of mRNA; the single nucleotide polymorphisms (SNPs) which compose the miRNA binding site in 3′-UTRs affect disease occurrence by regulating the affinity with miRNA [43,44,45]. In the present study, we investigated whether RPL risk is related to specific genetic polymorphisms in *SMAD5*, *RUNX-1*, and *FN3KRP* in Korean women. Specifically, we evaluated potential associations between RPL and *SMAD5* rs10515478 C>G, *FN3KRP* rs1046875 G>A, and *RUNX-1* rs15285 G>A, which are located in miRNA binding site-containing 3′-UTRs and have >5% minor allele frequencies in the East Asian population.

## 2. Materials and Methods

### 2.1. Study Population

We obtained blood samples from women with RPL seen at the CHA Bundang Medical Center’s Fertility Center (Seongnam, Korea) between March 1999 and February 2012. RPL was diagnosed based on human chorionic gonadotropin levels, ultrasonography, and physical examination prior to 20 weeks of gestation. Women were excluded if they had pregnancy loss caused by anatomic, hormonal, autoimmune, or thrombotic factors. Fetal anatomic abnormalities were identified by hysterosalpingogram, sonography, computerized tomography, hysteroscopy, or magnetic resonance imaging. Hormonal causes of RPL, such as luteal insufficiency, hyperprolactinemia, or thyroid disease, were evaluated by measuring levels of thyroid-stimulating hormone, prolactin, follicle-stimulating hormone, free T4, luteinizing hormone, and progesterone in peripheral blood. Autoimmune causes of RPL, such as lupus and antiphospholipid syndrome, were evaluated using lupus anticoagulant and anticardiolipin antibodies. Thrombotic causes of RPL were evaluated by deficiencies in protein C and protein S and the presence of anti-β2 glycoprotein antibodies. Control blood samples were obtained from healthy women seen at the CHA Bundang Medical Center who were confirmed to have a normal 46 XX karyotype, regular menstrual cycle, history of at least one naturally conceived pregnancy, and no history of pregnancy loss. In total, we analyzed 388 samples from women with RPL and 280 samples from healthy control women. No participants had a history of smoking or alcohol use, and all participants were Korean. This study was approved by the Institutional Review Board of CHA Bundang Medical Center (IRB number: BD2010-123D, 21 June 2011), and all participants provided written informed consent.

### 2.2. Genotype Analysis

Genomic DNA was extracted from whole blood by obtaining the buffy coat after centrifugation and employing the G DEX II Genomic DNA Extraction Kit (Intron Biotechnology Inc., Seongnam, Korea). Genetic polymorphisms were determined by genotype analysis. Real-time polymerase chain reaction (PCR) was performed for *SMAD5* rs10515478 C>G and *FN3KRP* rs1046875 G>A using the TaqMan SNP Genotyping Assay Kit (Applied Biosystems, Foster City, CA, USA). *RUNX-1* rs8134179 G>A was analyzed by PCR-restriction fragment length polymorphism analysis with digestion by the *Mnl*I restriction enzyme using forward primer 5′- GGC ACA GAG AAG GAG ATA TAG ACT -3′ and reverse primer 5′- ATA GTA TGC CAG GGC TCA GG -3′.

### 2.3. Assessment of Homocysteine, Folate, Total Cholesterol, and Uric Acid Concentrations and Blood Coagulation Status

Homocysteine levels were measured using a fluorescence polarization immunoassay and the Abbott IMx Analyzer (Abbott Laboratories, Abbott Park, IL, USA). Folate levels were measured using a radio-assay kit (ACS:180; Bayer, Tarrytown, NY, USA). Total cholesterol and uric acid levels were determined using commercially available enzymatic colorimetric tests (Roche Diagnostics, Mannheim, Germany). Platelets were measured using the Sysmex XE 2100 Automated Hematology System (Sysmex Corporation, Kobe, Japan). Prothrombin time and activated partial thromboplastin time were analyzed using an ACL TOP automated photo-optical coagulometer (Mitsubishi Chemical Medience, Tokyo, Japan).

### 2.4. Statistical Analysis

For data on participant characteristics, categorical variables were analyzed using Chi-square tests, and continuous variables were analyzed using Student’s *t*-tests. Multivariate logistic regression and Fisher’s exact tests were used to compare haplotype and genotype frequencies between RPL and healthy control women. Adjusted odds ratios (AORs) and 95% confidence intervals (CIs) were used to evaluate associations between genetic polymorphisms and RPL occurrence [46]. Model-based multifactor dimensionality reduction (MDR) was used to evaluate relationships between genotypes and participant characteristics [47,48,49]. Kruskal–Wallis tests were used for small sample sizes when the *p*-value of Levene’s test was less than 0.05. Statistical analysis was performed using GraphPad Prism 4.0 (GraphPad Software Inc., San Diego, CA, USA) and Medcalc version 12.7.1.0 (Medcalc Software, Mariakerke, Belgium). The HAPSTAT program (version3.0, www.bios.unc.edu/~lin/hapstat/, accessed on 17 March 2022) was used to evaluate the synergistic effects of polymorphic haplotypes. False discovery rate (FDR)-corrected *p*-values < 0.05 were considered to be statistically significant [50]. Data are reported as mean and standard deviation for continuous variables and frequencies and percentages for categorical variables.

## 3. Results

### 3.1. Participant Characteristics

We performed the age-matching of patients and controls and then undertook the comparison of clinical characteristics between two subjects. Compared with healthy control women, women with RPL had significantly shorter mean gestational durations and altered levels of pregnancy-related hormones (Table 1). Women with RPL also had significantly higher total cholesterol, blood urea nitrogen, platelets, prothrombin time, activated partial thromboplastin time, and hematocrit compared with healthy control women.

### 3.2. FN3KRP rs1046875 G>A Polymorphism Has Protective Role against RPL

Frequencies of *SMAD5* rs10515478 C>G, *FN3KRP* rs1046875 G>A, and *RUNX-1* rs15285 G>A polymorphisms were amplified and satisfied the Hardy–Weinberg principle (*p* > 0.05) on both control and patient groups (Table 2). We evaluated the association between each gene polymorphism and RPL and these results were adjusted by participant age. Among the three polymorphisms, only the *FN3KRP* rs1046875 AA genotype and recessive model differed significantly between women with RPL and healthy control women (Table 2). Specifically, women with RPL had a lower likelihood of the *FN3KRP* rs1046875 AA genotype and recessive model than healthy control women, and the protective role of rs1046875 AA genotype and recessive model became more pronounced when larger numbers of pregnancy losses (PL) had occurred (PL ≥ 2; AA genotype; AOR, 0.553; *p* = 0.010; recessive model; AOR, 0.631; *p* = 0.020; PL ≥ 3; AA genotype; AOR, 0.516; *p* = 0.016; recessive model; AOR, 0.594; *p* = 0.027); PL ≥ 4; AA genotype; AOR, 0.311; *p* = 0.009; recessive model; AOR, 0.333; *p* = 0.006). After FDR correction, the association between the *FN3KRP* rs1046875 AA genotype and RPL risk remained significant with all cases of PL but the recessive model remained significant only when PL ≥ 4.

### 3.3. Synergistic Effects of SMAD5, FN3KRP, and RUNX-1 Polymorphisms on RPL

Considering that RPL etiology has polygenic traits, combination analyses were performed to confirm associations with each allele or genotype combination and RPL. The likelihood of all combinations containing the A allele of *FN3KRP* rs1046875 G>A compared with the reference haplotype was lower among women with RPL than among healthy control women even though some were not statistically significant (Table 3). In particular, the combination of G, A, and G alleles in *SMAD5* rs10515478 C>G, *FN3KRP* rs1046875 G>A, and *RUNX-1* rs15285 G>A and of G and A alleles in *SMAD5* rs10515478 C>G and *FN3KRP* rs1046875 G>A were significantly less in women with RPL. After FDR correction, the G-A combination but not the G-A-G combination remained significantly associated with RPL. In addition, the likelihood of the combination of *SMAD5* rs10515478 CG and *FN3KRP* rs1046875 AA genotypes was significantly lower among women with RPL than among healthy control women (Table 4) holding the significance of allele combination.

### 3.4. Associations between Participant Characteristics and Genetic Polymorphisms

As women with RPL and healthy control women showed differences in clinicopathological factors (Table 1), we examined relationships between participant characteristics and each genetic polymorphism using MDR analysis. Both *FN3KRP* rs1046875 G>A (*p* = 0.046) and *RUNX-1* rs15285 G>A (*p* = 0.038) were significantly associated with platelet count, with the lowest level for hetero genotype (Table 5). In addition, *SMAD5* rs10515478 C>G was significantly associated with gestational duration with a longer duration for the GG genotype than for the CG or CC genotypes (CC, 18.45 ± 15.62; CG, 20.29 ± 15.64; GG, 25.82 ± 16.18).

## 4. Discussion

AGEs are products of the glycation of proteins or lipids resulting from exposure to sugars. High levels of AGEs are implicated in aging and the development or worsening of degenerative diseases including diabetes, atherosclerosis, chronic kidney disease, and Alzheimer’s disease [51]. The pathological contribution of AGEs to various diseases is under active investigation [4,52,53]. In particular, the accumulation of AGEs was found to contribute to sterility via damage to ovarian cells [31]. However, the role of AGEs in pregnancy loss has not previously been investigated. Therefore, we examined three genes previously linked to AGE formation and successful birth to determine their potential association with RPL. Furthermore, based on the contribution of non-coding regions to gene expression, regarding miRNA, we identified SNPs located in miRNA binding sites-containing 3′UTRs.

AGEs, which are produced by non-enzymatic glycation, have a deleterious impact on biological macromolecules via the formation of toxic compounds. However, the *FN3KRP* isoform of *FN3K* may protect proteins from non-enzymatic glycation [54]. Recent researches suggest that *FN3KRP* polymorphisms have protective effects against diabetes and cardiovascular diseases [26,28], which share common risk factors with pregnancy disorders such as vascular dysfunction and high glucose levels. Furthermore, a previous study reports increased *FN3KRP* expression when *FN3KRP* rs1046875 contains mutant alleles that alter its binding affinity with miR-34a [31]. As miR-34a is a negative regulator of angiogenesis and glucose metabolism [55], we hypothesize that *FN3KRP* rs1046875 mutations reduce levels of AGEs that can increase the risk of pregnancy loss (Figure 1). Consistent with previous studies, we found that women with RPL had a lower likelihood of a *FN3KRP* rs1046875 mutant genotype and recessive model, with stronger associations observed among women who experienced a larger number of pregnancy losses. These findings are the first to suggest the involvement of *FN3KRP* in RPL and possibly other pregnancy disorders.

Regarding allele combinations, we found that the G-A-G combination of *FN3KRP* rs1046875 G>A, *SMAD5* rs10515478 C>G, and *RUNX-1* rs15285 G>A was less likely in women with RPL than in healthy control women. *SMAD5* and *RUNX-1* are components of the TGF-β signaling pathway, which influences AGE formation (Figure 1). Specifically, *SMAD5* and *RUNX-1* appear to play similar roles in embryonic vascular development. A meshwork of blood vessels acts as an indispensable bridge for nutrient transmission from the mother to the fetus and is crucial for successful birth [7]. The importance of *SMAD5* and *RUNX-1* in pregnancy is confirmed by our finding of their interconnection with *FN3KRP*. Interestingly, the synergistic effect between *SMAD5* rs10515478 C>G and *FN3KRP* rs1046875 G>A maintained a significant association with decreased RPL occurrence in both allele and genotype combination analysis. Therefore we infer that an interaction between *FN3KRP* rs1046875 G>A and *SMAD5* rs10515478 C>G may reduce RPL risk and this interpretation is supported by previous reports that *SMAD5* but not *RUNX-1* influences AGEs directly through binding with TGF-β1.

Through additional analysis, we found association between *RUNX-1* rs15285 G>A and platelet count. Platelet formation is affected by high glucose levels, and platelet aggregation is reduced by insulin [56,57]. As a hematopoiesis gene, *RUNX-1* is believed to be closely linked to platelet count, with one study demonstrating that conditional knockout of *RUNX-1* in mice results in an 80% reduction in platelets [58]. Thus, *RUNX-1* rs15285 G>A may be the functional polymorphism underlying the involvement of *RUNX-1* in aspects of hematopoiesis for platelet formation, although this possibility requires further study. Unlike *RUNX-1*, there are no previous reports of a relationship between *FN3KRP* and platelets. However, we found that *FN3KRP* rs1046875 G>A was also associated with platelet count in Table 4. Based on the link between platelet formation and glucose concentration, our results suggest that *FN3KRP* rs1046875 G>A is involved in glucose metabolism and AGE formation, which should be confirmed by future research. In addition, *SMAD5* rs10515478 C>G was associated with gestational duration, consistent with our observation that mutant alleles of *SMAD5* rs10515478 C>G had protective effect against RPL when they occurred in combination with mutant alleles of *FN3KRP* rs1046875 G>A.

In summary, our results suggest that a *FN3KRP* rs1046875 G>A mutant genotype is associated with RPL risk. However, some limitations of our study should be considered. First, the mechanism by which each gene affects the development of RPL remains unclear; thus, future functional studies are required to unravel these processes. Second, the investigation of additional participant characteristics is needed to clarify the nature of the associations between genetic polymorphisms and RPL. Third, the population of our study was restricted to Korean women. To ensure that the investigated genetic polymorphisms can serve as biomarkers of RPL, studies involving more diverse ethnic populations are needed. Finally, larger sample sizes are required to more robustly support our conclusions.

## 5. Conclusions

In conclusion, we evaluated associations between *SMAD5* rs10515478 G>A, *FN3KRP* rs1046875 G>A, and *RUNX-1* rs15285 G>A polymorphisms and idiopathic RPL. To the best of our knowledge, this is the first study investigating the involvement of these SNPs with RPL risk. In particular, the role of *FN3KRP* in pregnancy disorders has not previously been explored. Here, we indicated the probability that the mutant genotype and recessive model of *FN3KRP* rs1046875 G>A may reduce the pregnancy loss, and its mutant allele may act synergistically with mutant allele of *SMAD5* rs10515478 G>A to protect against RPL. Furthermore, *FN3KRP* rs1046875 G>A and *RUNX-1* rs15285 G>A were associated with platelet count, which is linked to glucose concentration. Therefore, our study provides a valuable cornerstone for further *FN3KRP* research offering the hypothesis that *FN3KRP* may be associated with pregnancy loss regarding AGEs. The *FN3KRP* rs1046875 G>A polymorphism could serve as a therapeutic target for RPL by reducing HbA1c levels and thereby contribute to the lower level of AGEs, preventing pregnancy loss.

## Figures and Tables

**Figure 1 biomedicines-10-01481-f001:**
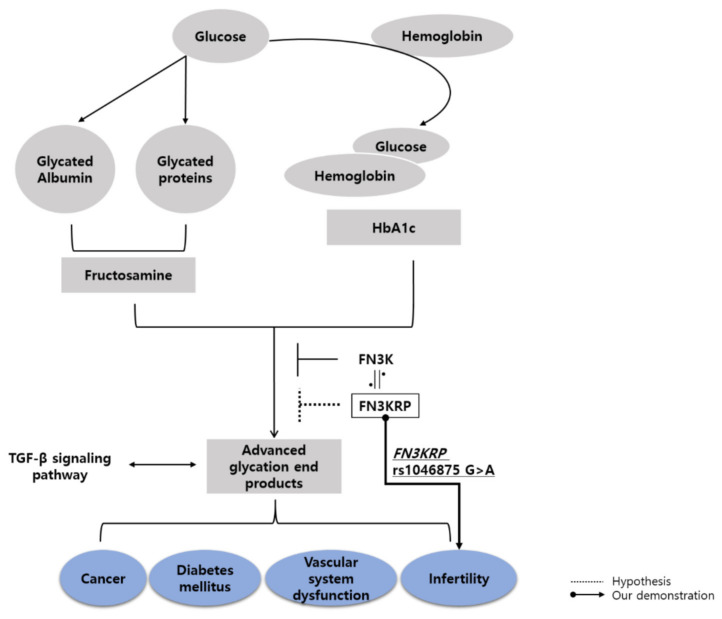
Overview of the formation and impact of AGEs. Proteins in the blood, such as albumin and hemoglobin, react with plasma glucose and generate glycated proteins including fructosamine and HbA1c, leading to the formation of AGEs through non-enzymatic glycation. The AGE formation process is restricted by *FN3K*, which is similar in structure to *FN3KRP.* The high level of AGEs can induce multiple diseases such as cancer, diabetes mellitus, vascular system dysfunction and infertility and interacts with the TGF- β signaling pathway. In our study, we indicated the association between *FN3KRP* rs1046875 G>A and RPL. Therefore, we propose that *FN3KRP* may inhibit the formation of AGEs and affect the RPL, infertility diseases.

**Table 1 biomedicines-10-01481-t001:** Clinical characteristics of control and RPL groups.

Characteristic	Control (*n* = 280)	RPL (*n* = 388)	*p **
Age (year, mean ± SD)	33.02 ± 5.74	33.21 ± 4.55	0.339 *
BMI (kg/m^2^, mean ± SD)	21.58 ± 3.18	21.49 ± 3.84	0.730 *
Previous pregnancy losses (*n*)	N/A	3.28 ± 1.84	
Live births (*n*)	1.72 ± 0.72	N/A	
Mean gestational age (weeks)	39.28 ± 1.67	7.36 ± 1.93	<0.0001 *
Homocysteine (µmol/L)	7.28 ± 1.58	6.98 ± 2.10	0.402 *
Folate (mg/mL)	13.71 ± 8.37	14.21 ± 11.94	0.887 *
Total cholesterol (mg/dL)	239.00 ± 85.19	187.73 ± 49.42	0.0004
Uric acid (mg/dL)	4.19 ± 1.44	3.80 ± 0.84	0.360 *
BUN (mg/dL)	8.03 ± 2.01	9.99 ± 2.77	<0.0001 *
Creatinine (mg/dL)	0.69 ± 0.08	0.72 ± 0.12	0.050 *
PLT (103/μL)	235.18 ± 63.60	255.43 ± 59.22	0.0007
PT (s)	11.53 ± 3.10	11.58 ± 0.86	<0.0001 *
aPTT (s)	30.78 ± 4.61	32.24 ± 4.33	0.006
Hct (%)	35.35 ± 4.26	37.31 ± 3.37	<0.0001 *
TSH (uIU/mL)	N/A	2.18 ± 1.55	
FSH (mIU/mL)	8.12 ± 2.85	7.52 ± 10.52	<0.0001 *
LH (mIU/mL)	3.32 ± 1.74	6.30 ± 12.09	<0.0001 *
E2 (Basal)	26.00 ± 14.75	35.71 ± 29.46	0.002 *
Prolactin (ng/mL)	N/A	15.68 ± 12.98	
FBS	N/A	95.24 ± 16.97	

Abbreviations: RPL, recurrent pregnancy loss; SD, standard deviation; N/A, not applicable; BMI, body mass index; BUN, blood urea nitrogen; PLT, platelets; PT, prothrombin time; aPTT, activated partial thromboplastin time; Hct, hematocrit; TSH, thyroid-stimulating hormone; FSH, follicle-stimulating hormone; LH, luteinizing hormone; E2, estradiol; FBS, fasting blood sugar. *p* *: *p*-values were calculated by chi-square test for categorical data and two-sided *t*-test for continuous data.

**Table 2 biomedicines-10-01481-t002:** Genotype frequencies of gene polymorphisms in control and RPL groups.

Genotype	Controls (*n* = 280)	PL ≥ 2 (*n* = 388)	AOR (95% CI)^c^	*p*	FDR-*p*	PL ≥ 3 (*n* = 206)	AOR (95% CI)	*p*	FDR-*p*	PL ≥ 4 (*n* = 82)	AOR (95% CI)	*p*	FDR-*p*
*SMAD5* rs10515478 C>G													
CC	92 (32.9)	137 (35.3)				69 (33.5)	1.000 (reference)			29 (35.4)	1.000 (reference)		
CG	138 (49.3)	194 (50.0)	0.949 (0.949–1.339)	0.766	0.766	100 (48.5)	0.965 (0.641–1.451)	0.863	0.954	35 (42.7)	0.808 (0.462–1.414)	0.456	0.920
GG	50 (17.9)	57 (14.7)	0.771 (0.771–1.225)	0.271	0.407	37 (18.0)	0.996 (0.587–1.691)	0.989	0.989	18 (22.0)	1.174 (0.589–2.340)	0.650	0.650
Dominant (CC vs. CG + GG)			0.894 (0.894–1.237)	0.499	0.499		0.965 (0.658–1.415)	0.854	0.903		0.890 (0.530–1.493)	0.658	0.954
Recessive (CC + CG vs. GG)			0.789 (0.789–1.197)	0.265	0.398		0.981 (0.612–1.572)	0.935	0.935		1.270 (0.687–2.348)	0.446	0.590
HWE-*p*	0.888	0.382											
*FN3KRP* rs1046875 G>A													
GG	70 (25.0)	122 (31.4)				65 (31.6)	1.000 (reference)			25 (30.5)	1.000 (reference)		
GA	141 (50.4)	200 (51.5)	0.820 (0.820–1.182)	0.288	0.687	108 (52.4)	0.819 (0.538–1.249)	0.354	0.954	49 (59.8)	0.972 (0.555–1.702)	0.920	0.920
AA	69 (24.6)	66 (17.0)	0.553 (0.553–0.866)	0.010	0.030	33 (16.0)	0.516 (0.302–0.882)	0.016	0.048	8 (9.8)	0.311 (0.130–0.742)	0.009	0.027
Dominant (GG vs. AG + AA)			0.733 (0.733–1.035)	0.078	0.234		0.723 (0.484–1.078)	0.111	0.333		0.761 (0.442–1.310)	0.324	0.954
Recessive (GG + AG vs. AA)			0.631 (0.631–0.922)	0.020	0.051		0.594 (0.374–0.943)	0.027	0.081		0.333 (0.153–0.725)	0.006	0.018
HWE-*p*	0.905	0.298											
*RUNX-1* rs15285 G>A													
GG	218 (77.9)	292 (75.3)				159 (77.2)	1.000 (reference)			64 (78.0)	1.000 (reference)		
GA	55 (19.6)	85 (21.9)	1.156 (1.156–1.694)	0.458	0.687	41 (19.9)	1.013 (0.643–1.596)	0.954	0.954	15 (18.3)	0.927 (0.491–1.751)	0.815	0.920
AA	7 (2.5)	11 (2.8)	1.167 (1.167–3.062)	0.753	0.753	6 (2.9)	1.136 (0.373–3.459)	0.823	0.989	3 (3.7)	1.423 (0.357–5.677)	0.618	0.650
Dominant (GG vs. AG + AA)			1.157 (1.157–1.666)	0.434	0.499		1.027 (0.667–1.582)	0.903	0.903		0.983 (0.542–1.782)	0.954	0.954
Recessive (GG + AG vs. AA)			1.135 (1.135–2.968)	0.796	0.796		1.140 (0.376–3.455)	0.816	0.935		1.460 (0.369–5.788)	0.590	0.590
HWE-*p*	0.128	0.122											

Note: The odds ratio was adjusted by age of participants. RPL, recurrent pregnancy loss; AOR, adjusted odds ratio; CI, confidence interval; HWE, Hardy–Weinberg equilibrium; FDR, False discovery rate.

**Table 3 biomedicines-10-01481-t003:** Haplotype analysis of gene polymorphisms in control and RPL groups.

Allele Combination	Controls (2*n* = 560)	RPL (2*n* = 776)	OR (95% CI)	*p*	FDR-*p*
*SMAD5* rs10515478 C>G/*FN3KRP* rs1046875 G>A/*RUNX-1* rs15285 G>A					
C-G-G	138 (24.6)	212 (27.4)	1.000 (reference)		
C-G-A	20 (3.6)	42 (5.4)	1.367 (0.770–2.427)	0.285	0.55
C-A-G	146 (26.0)	192 (24.7)	0.856 (0.632–1.160)	0.316	0.553
C-A-A	18 (3.3)	22 (2.9)	0.796 (0.412–1.538)	0.496	0.578
G-G-G	112 (20.0)	167 (21.5)	0.971 (0.704–1.339)	0.856	0.856
G-G-A	11 (2.0)	23 (3.0)	1.361 (0.643–2.881)	0.419	0.578
G-A-G	96 (17.1)	98 (12.7)	0.665 (0.467–0.947)	0.023	0.163
G-A-A	19 (3.5)	20 (2.5)	0.685 (0.353–1.331)	0.262	0.553
*SMAD5* rs10515478 C>G/*FN3KRP* rs1046875 G>A					
C-G	159 (28.3)	253 (32.7)	1.000 (reference)		
C-A	163 (29.2)	215 (27.7)	0.829 (0.624–1.102)	0.196	0.293
G-G	122 (21.9)	191 (24.6)	0.984 (0.728–1.330)	0.916	0.916
G-A	116 (20.6)	117 (15.1)	0.634 (0.458–0.877)	0.006	0.017
*SMAD5* rs10515478 C>G/*RUNX-1* rs15285 G>A					
C-G	284 (50.8)	404 (52.0)	1.000 (reference)		
C-A	38 (6.7)	64 (8.3)	1.184 (0.771–1.819)	0.440	0.660
G-G	207 (36.9)	265 (34.2)	0.900 (0.710–1.140)	0.383	0.660
G-A	31 (5.6)	43 (5.5)	0.975 (0.600–1.586)	0.919	0.919
*FN3KRP* rs1046875 G>A/*RUNX-1* rs15285 G>A					
G-G	250 (44.7)	379 (48.8)	1.000 (reference)		
G-A	31 (5.5)	65 (8.4)	1.383 (0.876–2.184)	0.163	0.184
A-G	241 (43.0)	290 (37.4)	0.794 (0.628–1.003)	0.053	0.158
A-A	38 (6.8)	42 (5.4)	0.729 (0.457–1.163)	0.184	0.184

Note: RPL, recurrent pregnancy loss; OR, odds ratio; CI, confidence interval; FDR, False discovery rate.

**Table 4 biomedicines-10-01481-t004:** Combination analysis of gene polymorphisms between control and RPL groups.

Genotype Combination	Controls (*n* = 280)	RPL (*n* = 388)	AOR (95% CI)	*p*	FDR-*p*
*SMAD5* rs10515478 C>G/*FN3KRP* rs1046875 G>A					
CC/GG	23 (8.2)	42 (10.8)	1.000 (reference)		
CC/GA	46 (16.4)	68 (17.5)	0.810 (0.431–1.523)	0.514	0.821
CC/AA	23 (8.2)	27 (7.0)	0.640 (0.300–1.365)	0.248	0.497
CG/GG	35 (12.5)	58 (14.9)	0.886 (0.457–1.716)	0.719	0.821
CG/GA	66 (23.6)	103 (26.5)	0.865 (0.476–1.572)	0.635	0.821
CG/AA	37 (13.2)	33 (8.5)	0.482 (0.241–0.967)	0.040	0.306
GG/GG	12 (4.3)	22 (5.7)	0.934 (0.387–2.255)	0.879	0.879
GG/GA	29 (10.4)	29 (7.5)	0.572 (0.275–1.190)	0.135	0.361
GG/AA	9 (3.2)	6 (1.5)	0.345 (0.106–1.120)	0.077	0.306
*SMAD5* rs10515478 C>G/*RUNX-1* rs15285 G>A					
CC/GG	71 (25.4)	103 (26.5)	1.000 (reference)		
CC/GA	20 (7.1)	30 (7.7)	0.999 (0.522–1.912)	0.999	0.999
CC/AA	1 (0.4)	4 (1.0)	3.196 (0.344–29.734)	0.307	0.859
CG/GG	109 (38.9)	147 (37.9)	0.923 (0.624–1.366)	0.688	0.999
CG/GA	25 (8.9)	41 (10.6)	1.162 (0.647–2.087)	0.616	0.999
CG/AA	4 (1.4)	6 (1.5)	0.960 (0.259–3.560)	0.951	0.999
GG/GG	38 (13.6)	42 (10.8)	0.763 (0.447–1.302)	0.322	0.859
GG/GA	10 (3.6)	14 (3.6)	0.963 (0.405–2.291)	0.932	0.999
GG/AA	2 (0.7)	1 (0.3)	0.291 (0.025–3.352)	0.322	0.859
*FN3KRP* rs1046875 G>A/*RUNX-1* rs15285 G>A					
GG/GG	54 (19.3)	95 (24.5)	1.000 (reference)		
GG/GA	16 (5.7)	24 (6.2)	0.875 (0.427–1.795)	0.716	0.832
GG/AA	0 (0.0)	3 (0.8)	NA	0.994	0.994
GA/GG	111 (39.6)	141 (36.3)	0.729 (0.480–1.107)	0.138	0.275
GA/GA	27 (9.6)	52 (13.4)	1.107 (0.624–1.964)	0.728	0.832
GA/AA	3 (1.1)	7 (1.8)	1.343 (0.333–5.414)	0.678	0.832
AA/GG	53 (18.9)	56 (14.4)	0.609 (0.368–1.008)	0.054	0.241
AA/GA	12 (4.3)	9 (2.3)	0.434 (0.171–1.098)	0.078	0.241
AA/AA	4 (1.4)	1 (0.3)	0.147 (0.016–1.352)	0.090	0.241

Note: The odds ratio was adjusted by age of participants. RPL, recurrent pregnancy loss; AOR, adjusted odds ratio; CI, confidence interval; N/A, not applicable; FDR, False discovery rate.

**Table 5 biomedicines-10-01481-t005:** Association between various clinical parameters and gene polymorphisms.

Genotype	BMI (kg/m^2^)	Previous Pregnancy Losses (*n*)	Mean Gestational Age (Weeks)	Hcy (umol/L)	Folate (ng/mL)	T.chol (mg/dL)	BUN (mg/dL)	Creatin (mg/dL)	PLT (10^3^/uL)	PT (s)	aPTT (s)
	Mean ± SD	Mean ± SD	Mean ± SD	Mean ± SD	Mean ± SD	Mean ± SD	Mean ± SD	Mean ± SD	Mean ± SD	Mean ± SD	Mean ± SD
*SMAD5* rs10515478 C>G											
CC	21.45 ± 3.08	2.99 ± 1.58	18.45 ± 15.62	7.12 ± 2.07	13.77 ± 11.13	194.84 ± 59.44	9.34 ± 2.78	0.71 ± 0.11	240.17 ± 60.58	11.70 ± 2.08	31.12 ± 4.64
CG	21.71 ± 4.27	2.94 ± 1.47	20.29 ± 15.64	6.92 ± 2.06	14.23 ± 12.51	187.54 ± 52.18	9.83 ± 2.83	0.72 ± 0.12	246.52 ± 59.86	11.39 ± 1.02	32.10 ± 4.44
GG	21.06 ± 2.57	3.33 ± 1.49	25.82 ± 16.18	6.90 ± 2.17	14.98 ± 9.76	198.36 ± 49.86	9.89 ± 2.37	0.73 ± 0.12	250.13 ± 73.55	11.90 ± 2.41	32.02 ± 4.07
*P* ^a^	0.374	0.222	0.023	0.722	0.885	0.546	0.382	0.512	0.480	0.155	0.184
*FN3KRP* rs1046875 G>A											
GG	21.76 ± 4.75	2.95 ± 1.28	19.56 ± 15.74	7.08 ± 2.13	14.22 ± 11.25	198.73 ± 59.39	9.69 ± 2.29	0.71 ± 0.11	256.76 ± 69.35	11.82 ± 2.42	32.13 ± 4.39
GA	21.33 ± 3.12	3.17 ± 1.78	20.86 ± 15.97	6.85 ± 1.81	13.51 ± 8.02	183.88 ± 42.74	9.45 ± 2.87	0.72 ± 0.12	239.37 ± 61.55	11.54 ± 1.33	31.80 ± 4.44
AA	21.67 ± 2.98	2.68 ± 0.86	21.94 ± 15.96	7.22 ± 2.63	16.29 ± 20.17	201.78 ± 71.12	10.11 ± 3.06	0.71 ± 0.13	242.67 ± 52.12	11.24 ± 0.96	30.96 ± 4.64
*P* ^a^	0.465	^b^ 0.376	0.675	0.462	^b^ 0.977	^b^ 0.470	0.373	0.905	0.046	0.155	0.274
*RUNX-1* rs15285 G>A											
GG	21.57 ± 3.86	3.05 ± 1.51	20.12 ± 15.78	6.87 ± 1.98	14.22 ± 12.16	189.80 ± 54.17	9.68 ± 2.72	0.72 ± 0.12	246.53 ± 61.83	11.59 ± 1.82	31.60 ± 4.51
GA	21.35 ± 3.03	2.87 ± 1.50	22.73 ± 16.25	7.30 ± 1.96	13.91 ± 10.32	202.42 ± 57.42	9.32 ± 2.67	0.70 ± 0.10	233.34 ± 60.17	11.54 ± 1.04	32.29 ± 4.26
AA	21.47 ± 2.65	3.27 ± 1.85	21.46 ± 16.19	8.07 ± 3.99	14.27 ± 7.36	167.75 ± 16.68	10.72 ± 4.58	0.70 ± 0.10	281.60 ± 80.81	11.30 ± 1.81	32.16 ± 5.20
*P* ^a^	0.853	0.541	0.507	^b^ 0.166	0.988	0.309	0.509	0.592	0.038	0.909	0.540

Note: BUN, blood urea nitrogen; T.chol, total cholesterol; BMI, body mass index; Hcy, homocysteine; PLT, platelet count; PT, prothrombin time; aPTT, activated partial thromboplastin time. ^a^ One-way analysis of variance test. ^b^ Kruskal–Wallis test.

## Data Availability

Publicly available datasets were analyzed in this study. This data can be found here: https://figshare.com/articles/dataset/Genetic_Polymorphisms_in_the_3_-Untranslated_Regions_of_SMAD5_FN3KRP_and_RUNX-1_Are_Associated_with_Recurrent_Pregnancy_Loss/20115293 (accessed on 27 April 2022).

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
