# Peer review of "Genetic Polymorphisms in the 3′-Untranslated Regions of SMAD5, FN3KRP, and RUNX-1 Are Associated with Recurrent Pregnancy Loss"

_biomedicines, 2022, doi:10.3390/biomedicines10071481_

Round 1

Reviewer 1 Report

Summary:

The manuscript focuses on associations between Recurrent pregnancy loss (RPL) risk and genetic polymorphisms of SMAD5, FN3KRP, and RUNX-1 in women of Korean ethnicity. Blood samples from 388 women with RPL and 280 healthy controls were studied for a specific period of time. RPL diagnosis was determined from human chorionic gonadotropin levels, ultrasonography, and physical examination before 20 weeks of gestation. Analysis were performed for genetic polymorphisms, homocysteine, folate, total cholesterol, uric acid concentrations, and blood coagulation status.

Data presented are:

Clinical characteristics, genotype frequencies, haplotype and combination analysis of gene polymorphisms, correlations between clinical parameters and gene polymorphisms

Strengths:

The study addresses a serious phenomenon that affects pregnancy whose cause is not well understood.

Various genotypic analysis are performed to decipher new insights into the condition

Large cohort was studied

Weaknesses:

Abstract: Advanced glycation end products (AGEs). In line 17, glycation is omitted.

Abstract conclusion needs to describe how the findings impact the future of the field.

Introduction: lines 33-34 – is the range 1-5% observed worldwide?

Results: Lines 67-70 and Figure 1: Is more information available to strengthen the context of HbA1c?

Figure 1: Representation and legend need to be edited.

Representation: Any entity that is a result, needs to be shown with arrow pointing to it. Any two interacting entities need to be connected with double-headed arrow.

Legend: All items represented in the figure need to be included in the legend.

Conclusion: line 132 – future applications need to be elaborated in 2-3 sentences.

Author Response

Thank you for dedicating your time to this manuscript revision.

We thank you for providing valuable feedback on our manuscript.

We revised some errors and awkward descriptions through revisions. The revised phrases or sentences are labeled by blue color.

We attached the PDF file, and also entered it below except picture.

Please find the revised manuscript.

Weaknesses:

Abstract: Advanced glycation end products (AGEs). In line 17, glycation is omitted.

Thank you for your comments and corrections. We added the glycation as follows.

SMAD5 and RUNX-1 are involved in the TGF-β signaling pathway and contribute to advanced glycation end products (AGEs) production and vascular development.” [Abstract, page 1, line 19]

Abstract conclusion needs to describe how the findings impact the future of the field.

Thank you for your comments. We agree with your comments and added the contents as you suggested considering that it is the first study of FN3KRP in pregnancy diseases.

“These findings suggest that the FN3KRP rs1046875 G>A polymorphism has a significant role on the prevalence of RPL in Korean women. Considering that it is the first study indicating the significant association between FN3KRP and pregnancy disease, RPL, our results shed the light to the further investigation of FN3KRP in infertility.” [Abstract, page 1, line 30-32]

Introduction: lines 33-34 – is the range 1-5% observed worldwide?

Thank you for your comments. Sorry for the confusion. Based on the review papers, we added the word “worldwide”.

“RPL occurs in 1-5% of reproductive-aged women worldwide, but its exact cause is unknown in approximately half of cases [2, 3].” [Introduction, page 1, line 38]

Results: Lines 67-70 and Figure 1: Is more information available to strengthen the context of HbA1c?

Thank you for your comments. We tried to augment some contents as you suggested and revised our insufficient descriptions as follows.

“In particular, FN3KRP rs1046875 G>A protects against cardiovascular diseases through HbA1c, glycated hemoglobin, and affects the binding affinity of miR-34a [31] which acts as a suppressor of angiogenesis and glucose metabolism [32, 33]. Although the mechanism of HbA1c has to be elucidated, its increased level has been reported in patients with cardiometabolic diseases and has positive correlation with the platelet dysfunction in diabetic mellitus [34-36]. As a precursor of hemoglobin-AGE, higher HbA1c concentration is also associated with various diseases such as vascular diseases, diabetes and pregnancy disorders [34, 36]. According to that, FN3KRP rs1046875 G>A has a potential probability to play a significant role in pregnancy regarding HbA1c.” [Introduction, page 2, line 69-78]

Figure 1: Representation and legend need to be edited.

Representation: Any entity that is a result, needs to be shown with arrow pointing to it. Any two interacting entities need to be connected with double-headed arrow.

Thank you for your comments. Based on your appreciable corrections, we changed the arrow and inserted the additional entities associated with our results.

Legend: All items represented in the figure need to be included in the legend.

Thank you for your comments. As you suggested, we added the contents of the represented diseases and added our results to clear the suggestion.

“The high level of AGEs can induce multiple diseases such as cancer, diabetes mellitus, vascular system dysfunction and infertility and interacts with the TGF- β signaling pathway. In our study, we indicated the association between FN3KRP rs1046875 G>A and RPL. Therefore, we propose that FN3KRP may inhibit the formation of AGEs and affect the RPL, infertility diseases.” [Figure 1, line 51-55]

Conclusion: line 132 – future applications need to be elaborated in 2-3 sentences.

Thank you for your comments. We agree that the contents should be supplemented. We appreciate your suggestion and have made the following corrections.

“Therefore, our study provides a valuable cornerstone for further FN3KRP research offering hypothesis that FN3KRP may be associated with pregnancy loss regarding AGEs and suggests that the FN3KRP rs1046875 G>A polymorphism could serve as a therapeutic target for RPL by reducing HbA1c levels thereby contributes to the lower level of AGEs, preventing the pregnancy loss.” [Conclusion, page 3, line 130-135]

Reviewer 2 Report

The manuscript entitled "Genetic polymorphisms in the 3’-untranslated regions of SMAD5, FN3KRP, and RUNX-1 are associated with recurrent pregnancy loss" is well written and diserve publishing in The Biomedicines. There are just few mistakes. In table 1 there is an error in the cholesterol value in Control group. The word "probability" is often used, could be replaced with "probability". The sentence ending in 55 line is lacking of citation. Thank you for superb work. 

Author Response

Thank you for dedicating your time to this manuscript revision.

We thank you for providing valuable feedback on our manuscript.

We revised some errors and awkward descriptions through revisions. The revised phrases or sentences are labeled by blue color.

We attached the PDF file, and also entered it below.

Please find the revised manuscript.

1. In table 1 there is an error in the cholesterol value in Control group.

Total cholesterol (mg/dL)

239.00±85.19

187.73±49.42

0.0004

Thank you for your comments. Sorry for the confusion. We corrected the mistake and also changed the p-value. We checked the other results related to the cholesterol value and there were no modifications.

2. The word "probability" is often used, could be replaced with "probability".

Thank you for your comments. Considering the importance of your suggestion, we used “probability” in two sentences as follows.

“According to that, FN3KRP rs1046875 G>A has a potential probability to play a significant role in pregnancy regarding HbA1c.” [Introduction, page 2, line 77]

“Here, we indicated that mutant genotype and recessive model of FN3KRP rs1046875 G>A has a probability to reduce the pregnancy loss, and its mutant allele may act synergistically with mutant allele of SMAD5 rs10515478 C>G to protect against RPL.” [Conclusion, page 2, line 127]

3. The sentence ending in 55 line is lacking of citation.

Thank you for your comments. We added the reference as follows.

“In addition, high blood sugar contributes to the formation of advanced glycation end-products (AGEs), which are toxic metabolic products of lipids, nucleic acids, and proteins formed by non-enzymatic reactions with sugars [9]. Elevated levels of AGEs are associated with severe health issues such as cancer and vascular diseases and may also play a role in infertility [4].” [Introduction, page 2, line 49, 51]

Reviewer 3 Report

Dear authors,

This manuscript provides relevant information related to polymorphisms of three genes related with recurrent pregnancy loss in women. The introduction provides sufficient background and includes relevant references,  the research design is appropriate, the methods are adequately described, the results are clearly presented, and the conclusions are supported by these results. The results are highly relevant in the study of genetic factors that affect embryonic loss in humans and I believe that the manuscript can be published in its current version.

Author Response

Thank you for dedicating your time to this manuscript revision.

We are grateful to your insightful and thoughtful comments on our paper.

Thank you for your helpful words and once again thank you for taking the time to revise.

Reviewer 4 Report

  1.     The description of the abstract is confusing to read. Please rewrite it.
  2.     Line 17: “advanced end products (AGEs)”: Please add ”glycation” within the term.
  3.     Line 52-53, Line 53-55: please cite the related reference
  4.   In Line 61: please add more information and studies regarding the SNPs of SMAD5 and RUNX-a and pregnancy loss.
  5.   The strength of association among angiogenesis, AGEs, and pregnancy loss was weak in the introduction section.
  6. Please indicate the sequence of all the primers used in this study and the PCR program.
  7.   In Table 1, the total cholesterol in the control group is 4.19, while that in the RPL group is 187. Is there some mistake or typing error in the data?
  8.     Based on the data in this manuscript, it could not be implied that FN3KPR inhibits the formation of AGEs shown in Figure 1. Figure 1 is not suitable to this manuscript.

Author Response

Thank you for dedicating your time to this manuscript revision.

We thank you for providing valuable feedback on our manuscript.

We revised some errors and awkward descriptions through revisions. The revised phrases or sentences are labeled by blue color.

We attached the PDF file, and also entered it below except picture.

Please find the revised manuscript.

 1.  The description of the abstract is confusing to read. Please rewrite it.

Thank you for your comments. Sorry for the confusion. Recognizing the awkward explanation in line 15, we tried to clarify the meaning of the association between vascular development, glucose and pregnancy loss by adding a few sentences. In addition, in order to clarify the significance of this study, the contents of specific research effects were added to future research fields.

“Although the causes of idiopathic RPL are not completely understood, vascular development and glucose concentration were reported to correlate with the pregnancy loss. The TGF-β signaling pathway which plays a significant role in pregnancy is activated by the interaction between high glucose and SMAD signaling, and affects the vascular cells.” [Abstract, page 1, line 14-17]

“These findings suggest that the FN3KRP rs1046875 G>A polymorphism has a significant role on the prevalence of RPL in Korean women. Considering that it is the first study indicating the significant association between FN3KRP and pregnancy disease, RPL, our results shed the light to the further investigation of FN3KRP in infertility.” [Abstract, page 1, line 29-32]

  2.  Line 17: “advanced end products (AGEs)”: Please add ”glycation” within the term.

Thank you for your comments and correction. We added the glycation as follows.

SMAD5 and RUNX-1 are involved in the TGF-β signaling pathway and contribute to advanced glycation end products (AGEs) production and vascular development.” [Abstract, page 1, line19]

   3.    Line 52-53, Line 53-55: please cite the related reference

Thank you for your comments. We agree with your comments and added the references.

[Introduction, page 2, line 47, 49 and 51]

 4. In Line 61: please add more information and studies regarding the SNPs of SMAD5 and RUNX-1 and pregnancy loss.

Thank you for your comments. We agree that the information of the SNPs should be complemented. However, there are none of studies that refer to the significant SNPs of SMAD5 and RUNX-1 in pregnancy loss while they are significant in vascular disease and platelet dysfunction. Therefore, we revised the manuscript as follows.

“However, genetic variants of SMAD5 and RUNX-1 were not investigated in pregnancy loss while there are reports about the significant polymorphisms in vascular disease and platelet dysfunction [24, 25].” [Introduction, page 2, line 62-64]

 5.  The strength of association among angiogenesis, AGEs, and pregnancy loss was weak in the introduction section.

Thank you for your comments. Sorry for the confusion and unclear content. For clarity, we have altered the content structure and added several sentences.

The changed structure: [Introduction, page 2, line 79-85]

The complemented sentences: [Introduction, page 2, line 45-47, 53-54, 59-60, 62-64 and 70-78]

6. Please indicate the sequence of all the primers used in this study and the PCR program.

Thank you for your comments. Sorry for missing detailed information. We added the sequence of forward and reverse primers used for PCR-RFLP.

RUNX-1 rs8134179 G>A was analyzed by PCR-restriction fragment length polymorphism analysis with digestion by the Mnl We restriction enzyme using forward primer 5’- GGC ACA GAG AAG GAG ATA TAG ACT -3’ and reverse primer 5’- ATA GTA TGC CAG GGC TCA GG -3’.” [Materials and Methods, page 3, line 122-123]

 7. In Table 1, the total cholesterol in the control group is 4.19, while that in the RPL group is 187. Is there some mistake or typing error in the data?

Thank you for your comments. Sorry for the confusion. We corrected the mistake and also changed the p-value. We checked the other results related to the cholesterol value and there were no modifications.

Total cholesterol (mg/dL)

239.00±85.19

187.73±49.42

0.0004

 8.  Based on the data in this manuscript, it could not be implied that FN3KPR inhibits the formation of AGEs shown in Figure 1. Figure 1 is not suitable to this manuscript.

Thank you for your comments. We agree that our study is insufficient to prove the contents of Figure 1. However, our results are consistent with the hypothesis proposed by other studies for FN3K and FN3KRP. To illustrate this point, we have inserted our results and previous proposals for FN3K and FN3KRP as some entities.
